# Revealing the Correlation between Altered Skin Lipids Composition and Skin Disorders

**Katerina Drakou** [1,*], **Andrea Tsianni** [1], **Faye Vrani** [1], **Valia Kefala** [2] **and Efstathios Rallis** [2]

1    Department of Aesthetics and Cosmetology, The Limassol College, Limassol 3075, Cyprus;
a.tsianni@tlc.ac.cy (A.T.); drfvrani@gmail.com (F.V.)
2    Department of Biomedical Sciences, School of Health Sciences and Welfare, University of West Attica,
122 43 Athens, Greece; valiakef@uniwa.gr (V.K.); efrall@otenet.gr (E.R.)
*    Correspondence: katerina.d@tlc.ac.cy

**Abstract:** Human skin layers serve as a barrier between the body and the environment, by preventing water loss and blocking the entry of chemicals, allergens, and microbes. Recent data showed that skin lipids are vital 'key players' of several functions and mechanisms performing in the skin, such as, barrier function and microbiome composition. Abnormalities in lipid composition have been observed in inflammatory cutaneous diseases with a disrupted skin barrier. This review aims to demonstrate the fundamental role of keratinocytes, sebocytes, and microbiome-derived lipids in the maintenance of the skin barrier. Furthermore, it would reveal the correlation between altered skin lipids' composition, microbiome, and the occurrence of certain dermatological disorders such as acne vulgaris, atopic dermatitis, psoriasis, and rosacea.

**Keywords:** skin lipids; acne vulgaris; atopic dermatitis; psoriasis; rosacea





## 1. Introduction

The skin is the largest organ of the human body with a surface area of approximately 1.7 square meters. It is a complex organ with many vital functions, such as protection, thermoregulation, sensation, immunity, and barrier [1]. It consists of layers, the outer layer is the epidermis, the middle layer is the dermis, and the deeper layer is the hypodermis (or subcutaneous tissue).

Balanced skin lipid composition is necessary for barrier integrity and stability, antimicrobial activity, and pH maintenance. Nowadays it is widely accepted that abnormalities in skin lipid composition is associated with several diseases such as acne vulgaris, rosacea, atopic dermatitis, and psoriasis [2–4]. Furthermore, microbiome diversity is a key mediator of skin wellness and health. Numerous studies have shown that disturbed microbiome is correlated with the emerge of several dermatological diseases [5].

In this review, we aim to clarify the correlation between altered lipid composition and disrupted microbiome with the occurrence and severity of the above cutaneous diseases.

## 2. Lipids of the Skin

Epidermis is a multilayered, lipid-rich epithelium that provides structural support and varies in thickness. It is classified into five sublayers: stratum corneum (SC) which is the outer layer of the skin, stratum lucidum (SL), stratum granulosum (SG), stratum spinosum (SS), and stratum basale (SB, the deepest layer). The cells of the epidermis include the keratinocytes (the predominant cell type), the melanocytes (that produce melanin which protects against ultraviolet radiation), the Langerhans cells (immune functions), and the Merkel cells (light touch sensation) [6].

The skin appendages are skin-associated structures including hair follicles, nails, and glands (sebaceous and sweat) [7]. The extracellular space of the epidermis is dominated by

lipids, mostly ceramides, acyl ceramides, cholesterol, cholesterol esters, and non-esterified fatty acids (NEFA) [4].

Dermis is a skin layer between the epidermis and the hypodermis. There are two anatomical regions that the dermis can be divided into, the papillary dermis, a thinner, superficial portion of the dermis that contains a relatively small and loose distribution of elastic and collagen fibrils within a significant amount of ground substance and the reticular dermis, which can extend up to approximately 25% by stretching the collagen fibers, while it can be squeezed, and the fibers are oriented horizontally. The main role of dermis is to support and protect the skin via thermoregulation and sensation, primarily due to the presence of a connective tissue framework [8,9]. The hypodermis (the deepest layer) is found between the dermis and the aponeurosis and fasciae of the muscles. It is recognized as an endocrine organ and immune response initiator [8–10].

The epidermal barrier is necessary for the protection of the skin by preventing the entrance of microorganisms, maintaining homeostasis and reducing water loss [11]. The average thickness of the SC ranges from 10 to 30 μm [12] and has a dual role in maintaining skin hydration. Externally, it has a water-repellent barrier due to keratinized cells, thus protecting against overfilling with water. Internally, it traps water due to natural moisturizing agents (NMF), to prevent water loss. The role of the 'guardian' is played by the NMF and the lipids that surround and protect them. Proper skin hydration maintains skin plasticity, enhances barrier function, protects skin from damages, and allows enzymes to act properly [13].

Lipids of the skin consist mainly of ceramides (50%), cholesterol and its esters (37%), cholesterol sulfate (2–5%), and NEFAs (10–15%). SC structure follows the model 'bricks and mortar' where the 'bricks' are keratin-rich keratinocytes held in place by the lipids which are the 'mortar' of the SC [14]. Ceramides (CERs) consist of a hydrophobic fatty acid chain linked to an amino-containing sphingoid base. There are 21 subclasses of CERs with different fatty acid chain length (Figure 1) [15]. Predominant CERs in normal epidermis are the (AH), (AP), (NH), and (NP). Long-chain CERs strongly involved in the maintenance of barrier function while the contribution of short-chain CERs in this function is negligible. CERs are bioactive metabolites implicated in epidermal regeneration and immune responses of the skin [11].

Cholesterol and its derivatives are lipophilic steroids with a stark, four ring (ABCD) structure. The main role of cholesterol is the preservation of flux and stability of the SC layer. Cholesterol sulfate seems to be implicated in the regulation of desquamation [4]. Squalene is a precursor of cholesterol, a polyunsaturated hydrocarbon, and an essential ingredient of sebum, secreted by sebaceous glands onto the skin. Squalene oxidation products seem to be involved in UV protection but also in comedogenic activity [16].

The majority of barrier NEFAs and fatty acids bound to CERs in the SC are saturated and unbranched, hydrocarbon chains length between C16–26. Fatty acids are implicated in the barrier function and in the maintenance of an acidic pH on the skin and are necessary for the antibacterial and the antimycotic function of the skin [4,14,17,18]. Triacylglycerols (TAGs) and diacylglycerols (DAGs) consist of glycerol and three fatty acids (triesters) and glycerol and two fatty acids (diesters), respectively. TAGs are important components of the sebum involved in the production of omega-O-acylceramides which are involved in proper structure formation of the membranes and skin permeability barrier [19].

## FATTY ACID CHAINS

| Ceramides classes | Non-hydroxy fatty acid (N) | α-Hydroxy fatty acid (A) | ω-Hydroxy fatty acid (O) | Esterified ω- hydroxyl fatty acid (EO) |
|---|---|---|---|---|
| Sphingosine (S) | (NS) | (AS) | (OS) | (EOS) |
| Phytosphingosine (P) | (NP) | (AP) | (OP) | (EOP) |
| 6- Hydroxysphingosine (H) | (NH) | (AH) | (OH) | (EOH) |
| Dihydrosphingosine (DS) | (NDS) | (ADS) | (ODS) | (EODS) |
| 4,14- Sphingadiene (SD) | (NSD) | (ASD) | (OSD) | (EOSD) |
| Dihydroxy-dihydrosphingosine or Dihydroxysphinganine (T) | (NT) | | | |

**Figure 1.** Nomenclature and structures of ceramide classes.

## 3. Lipids and Microbiome in Skin Diseases

### 3.1. Acne Vulgaris

Acne vulgaris is an inflammatory disease of the pilosebaceous unit mainly caused by sebum overproduction mediated primarily by hormones, hyperkeratinization of the hair follicles, abnormalities in sebum fatty acid composition, and altered bacterial colonization, leading to a disturbed microbiome [5,20,21]. It is characterized by the appearance of closed (whitehead) and open (blackhead) comedones, papules, pustules, and cysts. Acne is the most common cutaneous disease worldwide among adolescences, affecting 85–95% of girls and boys, however, it may frequently appear in adulthood as well [22].

Lipidomics analysis studies revealed a correlation between altered skin lipid composition and acne vulgaris [4]. Camera et al. showed extremely high levels of DAGs in sebum derived from juvenile acne patients compared to healthy subjects [23]. Increased levels of fatty acids and squalene were also presented in the same study [23]. Lipid composition is significantly different in male youth acne patients compared to control subjects. Fatty acids, unsaturated NEFAs, cholesterol, wax esters, squalene, and DAGs were increased in acne patients whereas, long chain saturated fatty acids, average ceramide chain length, saturated NEFAs, and linoleic acid were decreased in these patients [24]. In a recent study, higher levels of unsaturated NEFAs, squalene, TAGs, and shorter average ceramide chain length were reported in infant and adolescent acne patients compared to subjects with healthy skin [2]. In contrast with previous data, the same study showed an increase in the relative amount of CERs in infant acne patients [25]. Similar results were demonstrated in acne adolescents with dark skin [26]. TAGs, unsaturated NEFAs, wax esters, and squalene were elevated in the facial sebum of acne patients compared with non-acne subjects [26]. High quantity of TAGs seemed to promote acne occurrence and determined the severity in acne patients [27]. The higher the amount of TAGs was, the more severe the acne in the adolescents was [2]. Skin microbiome also plays a vital role in acne vulgaris. The main etiological mechanism of acne seems to be the hormone-dependent increased sebum production, which provides suitable living conditions to *Propionibacterium acnes* (*P. acnes*) [5]. Numerous studies have shown that *P. acnes* is the most abundant species of the microbiome predominantly colonized in the scalp and face of adolescents and adults below 50 years old [28]. *P. acnes* can hydrolyze triglycerides in sebum into free fatty acid and propionic acid, to secrete porphyrins which stimulate the oxidation of squalene, to increase the activity of diacylglycerol acyltransferase which promotes the composition of TAGs and to upregulate the expression of filaggrin. The abundance of these bio-products leads to the formation of comedones and to the increased proliferation and differentiation of keratinocytes resulting in hyperkeratinization [28–30].

High abundance of Firmicutes especially *Staphylococcus* genus and lower abundance of Proteobacteria was observed on the surface of comedones, pustules, and papules in acne patients, however, Actinobacteria (including Propionibacterium) and Bacteroidetes population remained the same at these areas [20]. A metagenomic study revealed that abundance of *P. acnes* was similar between acne patients and healthy subjects, whereas *P. acnes* strain population was significantly different; two strains of *P. acnes*, ribotype 004 and 005, were observed only in acne patients and not in healthy subjects [31].

### 3.2. Atopic Dermatitis

Atopic dermatitis (AD) is a chronic, inflammatory skin disease primarily triggered by defective skin barrier function and altered immune responses leading to clinical symptoms such as dry skin, eczema, and persistent itching.

Lipidomics analysis studies enhance our understanding about the correlation between abnormal lipid composition and etiopathogenesis of AD [32]. Emmert et al. showed an increase in short-chain CERs, free fatty acids, and cholesterol sulfate in AD patients compared to healthy individuals while the relative amount of long-chain CERs was reduced in these patients [33]. It was also revealed that three specific classes of CERs (NS, AS, and

ADS) were abundant in AD patients compared to healthy controls, whereas, the amount of all other ceramide classes was not changed between the two groups [33].

Schafer et al. reported a significant reduction in phospholipids, sterol esters, sphingolipids, CERs fraction, and an increase in free fatty acids and sterols in AD patients [34]. The increased free fatty acids can probably be explained by the existence of a unique enzyme that has been found in AD patients, called glucosylceramide/sphingomyelin deacylase. This enzyme activity leads to abnormal degradation of glycosylceramide into sphingosylphosphatidylcholine [35]. Several proteins, prosaposin, filaggrin, involucrin, cystatin A, and Ted-H-1 antigen, play an important role in barrier function and formation and were found to be decreased in AD patients [36].

Wang et al. showed a significant decrease of CERs, sphingomyelin, most triglycerides structures, and diglycerides in AD lesions in the infant group compared to the healthy group. These abnormalities in the lipid composition excessively affect the skin barrier function. Glycerophospholipids, specifically phosphatidylethanolamine and phosphatidic acid, were present in higher levels in AD infant patients compared to healthy infants. These two signaling molecules are crucial for the proper stability of the cell membrane, involved in cell apoptosis and cell proliferation, respectively [37]. Elevated levels of specific enzymes (lipoxygenases and cyclooxygenase) involved in polyunsaturated fatty acids (PUFAs) metabolism and subsequently in pro-inflammatory activity have been detected in AD patients [38]. Thin-layer chromatography results showed a reduction in squalene and wax esters levels in AD patients compared to healthy subjects [18,39]. In another study, serine protease activity in AD patients was found increased, leading to lipid reduction including CERs and interleukins IL-1a and IL-1b, a group of cytokines that trigger inflammation [40]. Due to defective barrier function noticed in AD patients, antigens easily access into the body leading to the increased production of T-helper type 2 (Th2) and subsequently to inflammation [40]. Th2 cytokines suppress the secretion of several antimicrobial proteins on the human skin, resulting in allowance for several different bacteria to expand in numbers [41,42].

Nowadays, the role of the microbiome in the etiopathogenesis of AD is undoubtedly critical and important. High throughput sequencing studies enhance our understanding of the altered population and the abnormal abundant colonization by various bacteria, observed in the skin of AD patients [43]. The preferred increase of abundance of Staphylococae species on AD lesional and non-lesional skin is well established [44]. *Staphylococcus aureus* (*S. aureus*) on the AD skin is highly increased and it is related to the severity of the AD disease by secreting several virulence factors [45,46]. Besides the abundance of *S. aureus* on the AD skin, increased colonization of *S. epidermidis*, *S. haemolyticus*, and *Corynebacterium bovis* was detected as well [47,48]. *Streptococcus*, *Propionibacterium*, *Acinetobacter*, and *Prevotella* colonization appear to be low in numbers on AD skin compared to that of healthy subjects [43]. Overall, results indicate that skin microbiome diversity is crucial for healthy skin and that in AD, diversity is significantly lower compared to healthy individuals [5].

### 3.3. Rosacea

Rosacea is a chronic inflammatory disease mainly affecting women above 30 years of age. Vascular dysfunction, altered quality of sebaceous lipid composition, and abnormal microbiome are primarily implicated in the pathogenesis of rosacea.

Rosacea is clinically characterized by centrofacial skin erythema, telangiectasias, papules, pustules, and phymas [49,50]. Although limited data regarding abnormalities of skin lipids in rosacea are reported, a study showed that the skin of patients with papulopustular rosacea produces increased amounts of fatty acids (C14:0) and low levels of the long chain saturated fatty acids (C20:0, C22:0, C23:0, C24:0) and monounsaturated fatty acid (C20:1) compared to healthy skin [3,51]. Interestingly, relative abundance of total cholesterol, triglycerides, esters, free fatty acids, and squalene on the skin of rosacea patients was not differed [52]. Further studies are needed to reveal the possible role of skin lipids composition in the pathogenesis of rosacea.

Increased levels of antimicrobial peptides secreted mostly by keratinocytes and enhanced secretion of serine proteases, causing inflammation, tissue damage, erythema, and telangiectasia on the skin of rosacea patients have been reported [53,54]. Microbiome diversity disturbances seem to play a critical role in pathophysiology of rosacea. *Demotex mites*, *S. epidermidis*, *Helicobacter pylori*, and *Bacillus olenorium* are found to be over-colonized on the skin of rosacea patients. These microorganisms induce the activation of inflammatory pathways. High abundance of Firmicutes, especially Staphylococcus genus, in erythematotelangiectatic rosacea was detected and Streptococcus genus in papulopustular rosacea, whereas lower abundance of Actinobacteria, mainly Propionibacterium, was detected compared to control subjects [55].

### 3.4. Psoriasis

Psoriasis is a chronic, immune-mediated skin disease causing erythematous, rough, scaling papules, and plaques. It has also been associated with obesity, cardiovascular diseases, depression, arthritis, hypertension, and inflammatory bowel disease [56]. It is characterized by hyperproliferation of keratinocytes, hyperkeratinization, and dermal inflammation, leading to disrupted skin barrier [57].

Recent lipidomics studies reveal an abnormal lipid composition in psoriatic patients [58]; Łuczaj et al. showed higher expression of several CERs in the keratinocytes (NS, NP, AS, ADS, AP, EOS) and fibroblasts (AS, ADS, EOS) of patients with psoriasis, whereas specific ceramide class NDS presented in lower levels compared to healthy controls [59]. Mathew et al. observed higher n-6/n-3 PUFA ratio in psoriasis patients compared to controls [60]. These abnormalities seem to affect fatty acid metabolism leading to induction of T-helper cell 17 (Th17) and consequently to inflammation [61]. Despite the abnormalities seen in surface skin lipids composition, plasma lipids concentration was also altered in patients with psoriasis [8,62].

Skin and gut microbiome diversity may also influence the occurrence of psoriasis. Recent metagenomic analysis data of gut microbiome revealed an increase of Firmicutes and Proteobacteria species, whereas, Bacteroidetes species presented in lower levels in patients with psoriasis compared to healthy individuals [63]. Gut Actinobacteria were unaffected in psoriasis and non-psoriasis individuals. Relative abundance of all principal phyla was lower in the gut microbiome of moderate to severe psoriasis patients compared to mild psoriasis [63]. In contrast, high throughput next-generation sequencing data of the skin microbiome showed that the overall quantity of Actinobacteria in psoriasis skin was significantly reduced compared to healthy controls [64]. Regarding the genus of microorganisms, psoriasis skin microbiome was decreased in Propionibacterium, Ethanoligenens, Macrococcus, Lactobacilli, and Burkholderia spp., whereas, it was enriched principally by Pseudomonas, Streptococcus, and Corynebacterium (psoriasis lesional skin) Conchiformibius, Lactococcus, Moraxella, and Acetobacter (psoriasis non-lesional skin) [61,65]. Tett et al. reported that Micrococcus species were abundant on the skin of psoriatic patients [66]. At species level, psoriasis non-lesional skin is mainly inhabited by *S. sciuri*, whereas, psoriasis lesional skin is over-colonized by *S. aureus, S. pettenkoferi, C. simulans* and *C. kroppenstedtii, Finegoldia,* and *Neisseriaceae* species. [61,65]. Malassezia, a genus of fungi, seems to be abundant in patients with psoriasis [66].

## 4. Conclusions

Several studies have demonstrated the central role of skin lipid composition in healthy skin. Lipidomics studies improve our understanding on how alteration of skin lipid composition can lead to several dermatological malfunctions. It seems that the skin barrier function is affected the most. These abnormalities are correlated with cutaneous diseases such as, acne vulgaris, atopic dermatitis, rosacea, and psoriasis. In addition, metagenomic analysis data enhance our knowledge of microbiome diversity and how this is affected in several skin diseases. To conclude, more studies are needed to clarify the role of skin lipids

composition and their possible association with skin microbiome and the pathogenesis of skin disorders.

**Author Contributions:** Conceptualization, V.K., K.D., and E.R.; validation, K.D., A.T. and F.V.; formal analysis, K.D. and A.T.; investigation, K.D.; data curation, K.D. and A.T.; writing—original draft preparation, K.D., A.T., V.K., F.V. and E.R.; writing—review and editing, V.K. and E.R.; supervision, V.K.; project administration, K.D. All authors have read and agreed to the published version of the manuscript.

**Funding:** This research received no external funding.

**Conflicts of Interest:** The authors declare no conflict of interest.

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
