# Peer review of "Revealing the Correlation between Altered Skin Lipids Composition and Skin Disorders"

_cosmetics, doi:10.3390/cosmetics8030088_

Round 1
Reviewer 1 Report
Very interesting work! Congrats!
Author Response
Thank you for your kind support.
Reviewer 2 Report
While the submitted manuscript tried to provide a brief summary of correlation between skin lipids and diseases (as described in the title of manuscript), unfortunately, reviewer thinks that it fails to deliver sufficient information to journal readers.
Most of all, reviewer cannot understand why authors tries to combine skin lipids and skin microbiota (exactly saying, the subjects discussed in the manuscript is microbiota, not microbiome) in this manuscript. There is no clear explanation connecting the changes of skin lipids and skin microbiota.
Also, even for the skin lipids part, there is no review but mere summary of previous reports. More extensive and intensive review requires as a review paper.
In section 2, Lipids in the three skin layers, reviewer thinks dermis and hypodermis are not proper themes of the submitted review, and authors should focus on stratum corneum lipids or skin surface lipids, not epidermal lipids.
Lastly, there many inappropriate terms and thorough revision by professional native speaker is strongly recommended.
Author Response
Dear reviewer,
I would like to thank you for your consideration of our Manuscript ID: cosmetics-1354774 entitled “Revealing the correlation between altered skin lipids composition and skin disorders”. Your comments help us to improve the manuscript. All correction were tracked in the revised text.
Point 1: While the submitted manuscript tried to provide a brief summary of correlation between skin lipids and diseases (as described in the title of manuscript), unfortunately, reviewer thinks that it fails to deliver sufficient information to journal readers.
Response 1: More information has been added
Point 2: Most of all, reviewer cannot understand why authors tries to combine skin lipids and skin microbiota (exactly saying, the subjects discussed in the manuscript is microbiota, not microbiome) in this manuscript. There is no clear explanation connecting the changes of skin lipids and skin microbiota.
Response 2: It was corrected
Point 3: Also, even for the skin lipids part, there is no review but mere summary of previous reports. More extensive and intensive review requires as a review paper.
Response 3: More extensive review has been done
Point 4: In section 2, Lipids in the three skin layers, reviewer thinks dermis and hypodermis are not proper themes of the submitted review, and authors should focus on stratum corneum lipids or skin surface lipids, not epidermal lipids.
Response 4: It was corrected
Point 5: Lastly, there many inappropriate terms and thorough revision by professional native speaker is strongly recommended.
Response 5: Extensive editing of English language and style have been done
Reviewer 3 Report
A nice and comprehensive review, interesting review on the role of barrier function and barrier lipids on the maintenance of healthy skin. New aspects covering microbiome and new areas of recent research are covered
Author Response
Thank you for your kind support.
Round 2
Reviewer 2 Report
While the submitted manuscript improved much, there are still many points of revision before being accepted for publication.
For the discussion of ceramides, it is strongly recommended to add figures showing the structural features of ceramides, including nomenclature as well. In line 76 of page 2, "21 subclasses of CERS" is commented, but there is no reference supporting it.
In line 149 of page 4, "three certain classes of CERs" are commented, but it should be clarified in detail.
In line 156 of page 4, currently identified roles of "sphingomyelin deacylase" is abnormal degradation of glycosylceramide into sphingosylphosphatidylcholine, not disruption of free fatty acids.
In line 161 of page 4, what is the meaning of "(8 out of 9)"?
Also, there are many inappropriate expressions needs to be revised and clarified.
Author Response
Point 1: For the discussion of ceramides, it is strongly recommended to add figures showing the structural features of ceramides, including nomenclature as well. In line 76 of page 2, "21 subclasses of CERS" is commented, but there is no reference supporting it.
Response 1: Reference and figure have been added
Point 2: In line 149 of page 4, "three certain classes of CERs" are commented, but it should be clarified in detail.
Response 2: classes of CERs were added
Point 3: In line 156 of page 4, currently identified roles of "sphingomyelin deacylase" is abnormal degradation of glycosylceramide into sphingosylphosphatidylcholine, not disruption of free fatty acids.
Response 3: It was corrected
Point 4: In line 161 of page 4, what is the meaning of "(8 out of 9)"?
Response 4: It was corrected
Point 5: Also, there are many inappropriate expressions needs to be revised and clarified.
Response 5: Moderate editing of English language and style have been done
Round 3
Reviewer 2 Report
The manuscript was revised well and reviewer does not have any further comments.